# Upper airway gene expression reveals suppressed immune responses to SARS-CoV-2 compared with other respiratory viruses

Eran Mick [1,2,3,6], Jack Kamm [3,6], Angela Oliveira Pisco[3], Kalani Ratnasiri [3], Jennifer M. Babik[1], Gloria Castañeda[3], Joseph L. DeRisi[3,4], Angela M. Detweiler[3], Samantha L. Hao [3], Kirsten N. Kangelaris[5], G. Renuka Kumar[3], Lucy M. Li [3], Sabrina A. Mann[3,4], Norma Neff [3], Priya A. Prasad[5], Paula Hayakawa Serpa[1,3], Sachin J. Shah[5], Natasha Spottiswoode[5], Michelle Tan[3], Carolyn S. Calfee[2], Stephanie A. Christenson[2], Amy Kistler[3,7] & Charles Langelier [1,3,7 ✉]

SARS-CoV-2 infection is characterized by peak viral load in the upper airway prior to or at the time of symptom onset, an unusual feature that has enabled widespread transmission of the virus and precipitated a global pandemic. How SARS-CoV-2 is able to achieve high titer in the absence of symptoms remains unclear. Here, we examine the upper airway host transcriptional response in patients with COVID-19 ($n = 93$), other viral ($n = 41$) or non-viral ($n = 100$) acute respiratory illnesses (ARIs). Compared with other viral ARIs, COVID-19 is characterized by a pronounced interferon response but attenuated activation of other innate immune pathways, including toll-like receptor, interleukin and chemokine signaling. The IL-1 and NLRP3 inflammasome pathways are markedly less responsive to SARS-CoV-2, commensurate with a signature of diminished neutrophil and macrophage recruitment. This pattern resembles previously described distinctions between symptomatic and asymptomatic viral infections and may partly explain the propensity for pre-symptomatic transmission in COVID-19. We further use machine learning to build 27-, 10- and 3-gene classifiers that differentiate COVID-19 from other ARIs with AUROCs of 0.981, 0.954 and 0.885, respectively. Classifier performance is stable across a wide range of viral load, suggesting utility in mitigating false positive or false negative results of direct SARS-CoV-2 tests.

---

[1] Division of Infectious Diseases, University of California, San Francisco, CA, USA. [2] Division of Pulmonary and Critical Care Medicine, University of California, San Francisco, CA, USA. [3] Chan Zuckerberg Biohub, San Francisco, CA, USA. [4] Department of Biochemistry and Biophysics, University of California, San Francisco, CA, USA. [5] Division of Hospital Medicine, University of California, San Francisco, CA, USA. [6] These authors contributed equally: Eran Mick, Jack Kamm. [8] These authors jointly supervised this work: Amy Kistler, Charles Langelier. ✉email: chaz.langelier@ucsf.edu

The emergence of severe acute respiratory syndrome coronavirus 2 (SARS-CoV-2) in December 2019 has precipitated a global pandemic with over 45 million cases and 1 million deaths[1]. Coronavirus disease 2019 (COVID-19), the clinical syndrome caused by SARS-CoV-2, is characterized by peak viral load and transmissibility prior to or at the time of symptom onset[2–8] with a disease course that varies from asymptomatic infection to critical illness[9]. Defining the host response to SARS-CoV-2, as compared to other respiratory viruses, is fundamental to identifying mechanisms of pathogenicity and potential therapeutic targets.

Metagenomic next-generation RNA sequencing (mNGS) is a powerful tool for assessing host/pathogen dynamics and a promising modality for developing novel respiratory diagnostics that integrate host transcriptional signatures of infection[10,11]. While proven for diagnosis of other acute respiratory infections, transcriptional profiling has not yet been evaluated as a diagnostic tool for COVID-19, despite its potential to mitigate the risk of false-positive or false-negative outcomes associated with standard reverse transcription-polymerase chain reaction (RT-PCR) testing for viral RNA from nasopharyngeal/oropharyngeal (NP/OP) swabs[12–14].

Here, we apply mNGS to examine the upper airway host transcriptional response in patients with COVID-19, other viral or non-viral acute respiratory illnesses (ARIs). We find that COVID-19 is characterized by markedly attenuated activation of innate immune and pro-inflammatory pathways early in the course of disease compared to other viral ARIs, which may partly explain the propensity for pre-symptomatic transmission of SARS-CoV-2. We further develop parsimonious classifiers based on patient gene expression that accurately differentiate COVID-19 from other ARIs.

## Results

To interrogate the molecular pathogenesis of SARS-CoV-2 and evaluate the feasibility of a COVID-19 diagnostic based on host gene expression, we conducted a multicenter observational study of 234 patients with ARIs who were tested for SARS-CoV-2 by NP/OP swab RT-PCR, and performed mNGS on the same swab specimens. The cohort (Supplementary Table 1) included: (i) 93 patients who tested positive for SARS-CoV-2 by PCR early in the course of disease and had no other pathogenic respiratory virus detected by mNGS, (ii) 41 patients who tested negative for SARS-CoV-2 but had another respiratory virus detected by mNGS (Methods; Supplementary Fig. 1a), and (iii) 100 patients who tested negative for SARS-CoV-2 and had no other virus detected by mNGS (non-viral ARIs). Diagnoses in the latter group included bacterial pneumonia and non-infectious lung and airway conditions, though a definitive etiology could not be determined in every case (Supplementary Data 1).

We began by performing pairwise differential expression (DE) analyses between the three patient groups (Methods; Supplementary Data 2). Hierarchical clustering of the union of the 50 most significant genes in each of the comparisons revealed commonalities and distinctions in the transcriptional response to SARS-CoV-2 and other viruses (Fig. 1a). Many genes were upregulated in all viral ARIs and appeared to be induced proportionally to SARS-CoV-2 viral load, as measured by the relative abundance of sequencing reads mapped to the virus (Methods; Supplementary Fig. 1b). However, we also detected gene clusters that were up- or down-regulated by other viruses as compared to non-viral ARIs that remained relatively unaffected by SARS-CoV-2. Only few genes were upregulated by SARS-CoV-2 more than by other viruses.

To investigate the pathways driving these patterns, we performed gene set enrichment analyses[15] (GSEA) on the genes differentially expressed (DE) between SARS-CoV-2 and non-viral ARIs, and separately, on the genes DE between other viral ARIs and non-viral ARIs (Methods; Supplementary Data 3). We found that both SARS-CoV-2 and other viruses elicited a robust interferon response in the upper airway (Fig. 1b). The most statistically significant genes upregulated by SARS-CoV-2 were interferon inducible, including *IFI6, IFI44L, IFI27, IFI44, HERC6, OAS2,* and *IFIT1,* in general agreement with previous reports[16,17]. *IFI27* and, to a lesser degree, *IFI6,* were induced by SARS-CoV-2 more than by other viruses but most top DE genes did not specifically distinguish SARS-CoV-2 (Supplementary Figs. 2a, 3a). *ACE2,* which encodes the cellular receptor for SARS-CoV-2, also appeared to be non-specifically induced, consistent with its identification as a general interferon-stimulated gene[18] (Supplementary Figs. 2a, 3a). However, recent reports suggest this signal is driven by an *ACE2* isoform that lacks the viral binding domains and is unlikely to encode a functional receptor[19,20].

Nevertheless, GSEA of DE genes in the direct comparison of SARS-CoV-2 and other viruses suggested some elements of the interferon response to SARS-CoV-2 were attenuated (Supplementary Fig. 2b; Supplementary Data 3). Indeed, several interferon response genes, such as *IRF7* and *OASL,* were more potently induced by other viruses, and high SARS-CoV-2 viral load was required to achieve comparable induction (Supplementary Fig. 2c). These results may be related to observations of a blunted interferon response in cellular models of SARS-CoV-2 infection[21], though the effects in patients are considerably more nuanced.

A striking contrast between SARS-CoV-2 and other viruses emerged in the activation of additional innate immune signaling pathways. Other viruses caused significant upregulation of gene expression associated with toll-like receptors, interleukin signaling, chemokine binding, inflammasomes, neutrophil degranulation and interactions with lymphoid cells, yet the response of these pathways to SARS-CoV-2 was markedly attenuated (Fig. 1b, Supplementary Fig. 2b). Other viruses depressed expression of genes involved in cilia functions as well as ribosomal protein genes and certain mitochondrial functions, which was not observed for SARS-CoV-2, whereas SARS-CoV-2 specifically depressed expression of olfactory receptors, consistent with the loss of sense of smell frequently reported in COVID-19[22,23] (Fig. 1b, Supplementary Fig. 2b).

Certain differences in gene expression between patient groups may be driven by changes in tissue cellular composition, including through recruitment of immune cell types to the site of infection. To examine this, we performed in silico estimation of cell-type proportions in the bulk RNA sequencing data using markers previously derived from airway single-cell sequencing studies (Methods; Supplementary Data 4). Strikingly, while patients with other viral ARIs exhibited significantly increased proportions of monocytes/macrophages and neutrophils in the upper airway compared to non-viral ARIs, this was not the case for those infected with SARS-CoV-2 (Fig. 1c, Supplementary Fig. 4). SARS-CoV-2 infection increased proportions of dendritic and B-cells more than other viruses, while other viruses decreased proportions of ciliated cells and goblet cells. These results closely aligned with the GSEA findings and suggested that the diminished innate immune responses observed in COVID-19 patients were coupled to blunted recruitment and activation of pro-inflammatory macrophages in the airway microenvironment.

Further supporting these findings, we found that the gene most depressed in expression in COVID-19 patients compared to those with other viral ARIs was *IL1B,* which encodes a pro-inflammatory cytokine produced by the inflammasome complex,

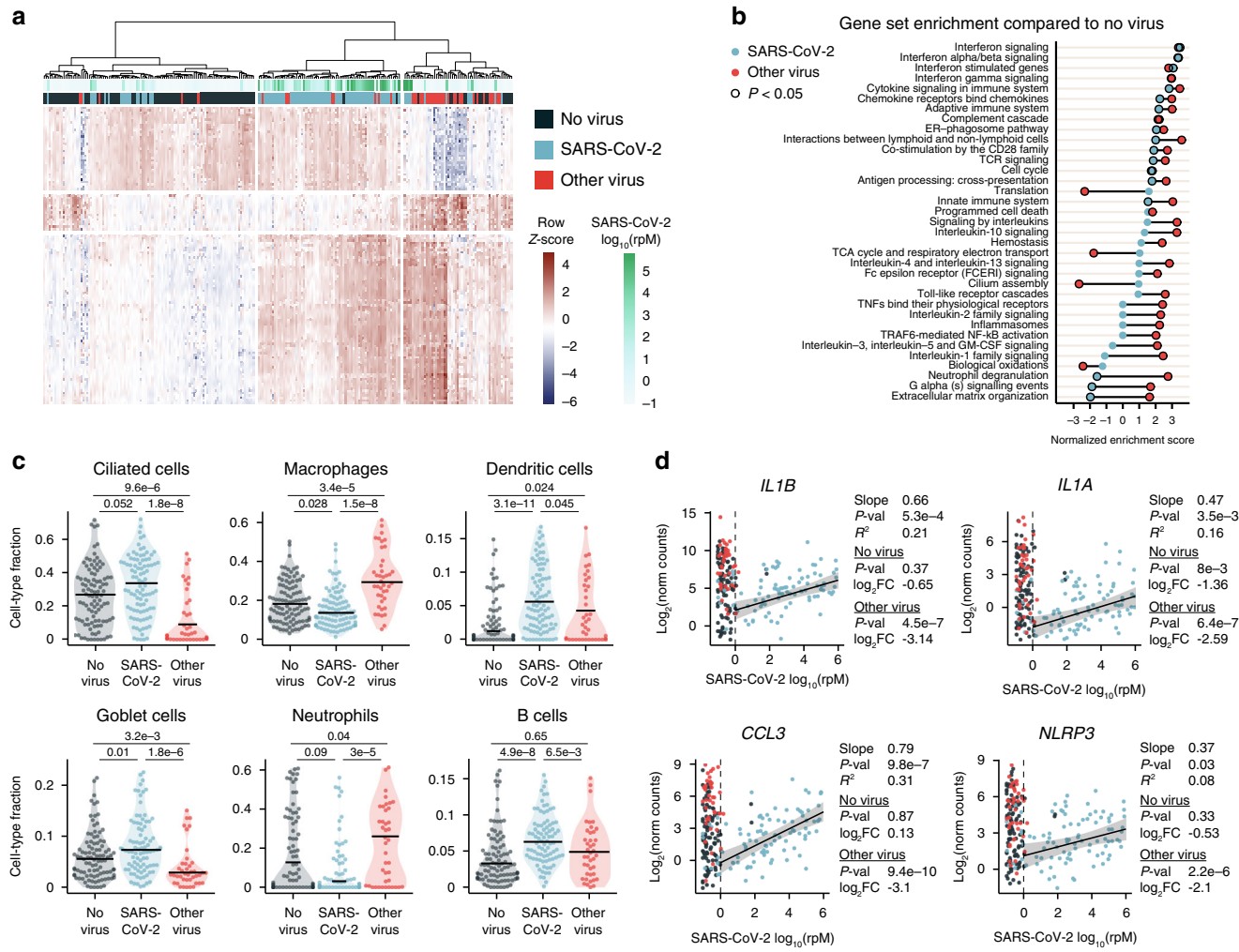

**Fig. 1 Host Transcriptional Signatures of SARS-CoV-2 Infection as Compared to Other Respiratory Viruses. a** Hierarchical clustering of 121 genes comprising the union of the top 50 differentially expressed (DE) genes by significance in each of the pairwise comparisons between patients with COVID-19 (SARS-CoV-2; $n = 93$), other viral ARIs ($n = 41$) and non-viral ARIs ($n = 100$). Gene expression values were clustered after variance-stabilizing transformation and row normalization. Group labels and viral load of SARS-CoV-2 are shown in the annotation bars. rpM, reads-per-million. **b** Normalized enrichment scores of selected REACTOME pathways that achieved statistical significance in at least one of the gene set enrichment analyses, using either DE genes between SARS-CoV-2 and non-viral ARIs or between other viral ARIs and non-viral ARIs. If a pathway could not be tested in one of the comparisons since it had <10 members in the input gene set, the enrichment score was set to 0. Pathway p-values were calculated using an adaptive, multilevel splitting Monte Carlo approach and Benjamini–Hochberg adjusted. **c** In silico estimation of cell-type proportions in the bulk RNA sequencing using single-cell signatures. Black lines denote the median. The y-axis in each panel was trimmed at the maximum value among the three patient groups of 1.5*IQR above the third quartile, where IQR is the inter-quartile range. Pairwise comparisons between patient groups were performed with a two-sided Mann–Whitney–Wilcoxon test followed by Bonferroni's correction. Sample sizes as in (**a**). **d** Scatter plots of normalized gene counts (log$_2$ scale, y-axis) as a function of SARS-CoV-2 viral load (log$_{10}$(rpM), x-axis). Shown are inflammasome-related genes selected from among the genes most depressed in expression in SARS-CoV-2 compared to other viral ARIs. Robust regression was performed on SARS-CoV-2 positive patients with log$_{10}$(rpM) $\geq 0$ ($n = 82$) to characterize the relationship to viral load. Shaded bands represent 95% confidence intervals. Statistical results listed for each gene refer to, from top to bottom: the regression analysis (p-values for difference of the slope from 0 derived from a t-statistic and Benjamini–Hochberg adjusted; R$^2$ is the adjusted robust coefficient of determination), the DE analysis between SARS-CoV-2 and non-viral ARIs (p-values derived from a moderated t-statistic and Benjamini–Hochberg adjusted), and the DE analysis between SARS-CoV-2 and other viral ARIs (p-values derived from a moderated t-statistic and Benjamini–Hochberg adjusted). Sample sizes for DE analyses as in (**a**). FC, fold-change.

particularly in macrophages[24,25] (Fig. 1d, Supplementary Data 2). Moreover, among the top 100 differentially decreased genes were those involved in inflammasome activation and activity[26] (*NLRP3, CASP5, IL1A, IL1B, IL18RAP,* and *IL1R2*) and in chemokine signaling for recruiting innate immune cells to the epithelium (*CCL2, CCL3,* and *CCL4*). Importantly, we recapitulated these findings in a re-analysis we performed on a large, independent dataset of NP swab mNGS that included 166 patients with COVID-19 and 79 patients with other viral ARIs[16,27] (Methods; Supplementary Data 2). We note that the muted

response of the IL-1 and inflammasome pathways to SARS-CoV-2 infection appeared to distinguish it from most other pathogenic respiratory viruses in our cohort, including common cold coronaviruses, with the possible exception of influenza (Supplementary Fig. 3b).

Relatively few genes were specifically upregulated in COVID-19 patients compared to both other viral and non-viral ARIs. These included *TRO,* which encodes a membrane-bound cell adhesion molecule; *WDR74,* which plays a role in rRNA processing and associates with the RNA helicase MTR4[28]; *EIF4A2,* a

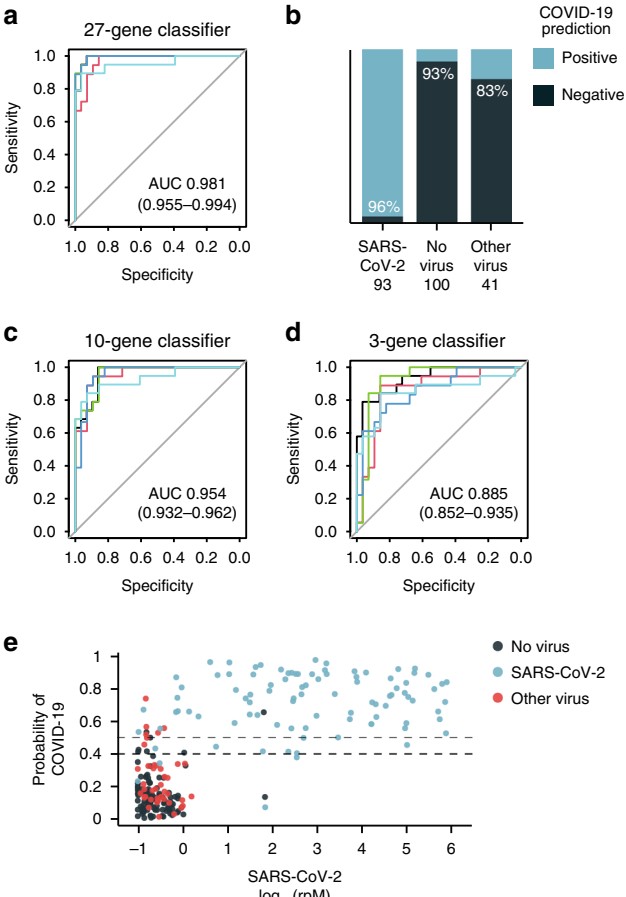

**Fig. 2 Performance of COVID-19 diagnostic classifiers based on patient gene expression. a** Receiver operating characteristic (ROC) curve for a 27-gene classifier that differentiates COVID-19 from other acute respiratory illnesses (viral and non-viral). The mean and range of the area under the curve (AUC) are indicated. **b** Accuracy of the 27-gene classifier within each patient group using a cut-off of 40% out-of-fold predicted probability for COVID-19. **c** ROC curve for a 10-gene classifier. **d** ROC curve for a 3-gene classifier. **e** Out-of-fold predicted probability of COVID-19 derived from the 27-gene classifier plotted as a function of SARS-CoV-2 viral load, $\log_{10}$(rpM). Dashed lines indicate 40% (our chosen cut-off) and 50%.

translation initiation factor that has been shown to interact with other coronaviruses as well as HIV[29,30]; and *FAM83A*, which is involved in epidermal growth factor receptor (EGFR) signaling[31] (Supplementary Fig. 3c).

We next asked whether host gene expression data could be used to construct a classifier capable of accurately differentiating COVID-19 from other ARIs (viral or non-viral). By employing a combination of lasso regularized regression and random forest (Methods), we first identified a 27-gene signature that performed with an area under the receiver operating characteristic curve (AUROC) of 0.981 (range of 0.955–0.994), as estimated by five-fold cross validation (Fig. 2a, Supplementary Tables 2, 3). Even though many patients undergoing testing for COVID-19 may not be infected with other respiratory viruses, we recognized the need for classifier specificity in this circumstance and examined how well the classifier performed at distinguishing SARS-CoV-2 from other respiratory viruses. We found that it achieved an AUROC of 0.966 (range 0.921–1.000) when tested only on patients with other viral ARIs, indicating robust specificity for SARS-CoV-2 (Supplementary Tables 2, 3). Using a cut-off of 40% predicted

out-of-fold probability for COVID-19 to call a patient positive, this translated into a sensitivity of 96% and a specificity of 93% for patients with non-viral ARIs and 83% for patients with other viral ARIs (Fig. 2b).

Given that a parsimonious gene set could enable practical incorporation into a clinical PCR assay, we implemented a more restrictive regression penalty and identified a 10-gene classifier that could differentiate SARS-CoV-2 from other respiratory illnesses with an AUROC of 0.954 (range 0.932–0.962) (Fig. 2c; Supplementary Tables 2, 3). Classification performance specifically against other viral ARIs suffered slightly but still achieved an AUROC of 0.912 (range 0.868–0.947). Existing SARS-CoV-2 PCR assays typically employ 3 gene targets and thus we tested the potential to further reduce host classifier gene size. We found that a sparse 3-gene (*IL1B, IFI6,* and *IL1R2*) classifier still achieved an AUROC of 0.885 (range 0.852–0.935) (Fig. 2d; Supplementary Tables 2, 3).

A host-based diagnostic might have particular utility if it could increase the sensitivity of standard NP/OP swab PCR testing, which may return falsely negative in a significant proportion of patients[12–14]. Presumably, false negatives are in large part due to insufficient viral abundance in the collected specimen. While our cohort did not include PCR-negative samples from patients with clinically confirmed COVID-19, we evaluated whether classifier performance was affected by viral load. The predicted probability of SARS-CoV-2 infection had little apparent relationship to the abundance of SARS-CoV-2, suggesting host gene expression has the potential to provide an orthogonal indication of COVID-19 status even when viral abundance is low (Fig. 2e).

## Discussion

We studied patients with ARIs to define the human upper airway gene expression signature in COVID-19. Our results reveal an attenuated innate immune response to SARS-CoV-2 as compared to other respiratory viruses. The IL-1 and NLRP3 inflammasome pathways were particularly non-responsive to SARS-CoV-2, commensurate with a signature of impaired neutrophil and macrophage recruitment to the upper airway, at least early in the course of the disease.

The blunted activation of these pro-inflammatory pathways in the upper airway carries two important implications. First, it suggests a potential mechanism underlying the observation of high viral titers in the upper airway prior to symptom onset in infected individuals, which poses a major challenge to preventing viral transmission in the COVID-19 pandemic[32,33] and distinguishes it from the SARS outbreak of 2002-2003[34]. Intriguingly, a human challenge study with influenza virus reported that blunted activation of the IL-1 and NLRP3 inflammasome pathways was specifically associated with an asymptomatic course of infection[35], supporting this connection. Influenza was indeed the only other virus that showed a degree of similarity to SARS-CoV-2 in this regard in our data. Future work is needed to identify potential SARS-CoV-2 factors responsible for this immune attenuation and to clarify the underlying biological mechanisms.

Second, given that IL1-β and other pro-inflammatory cytokines are primary targets of monoclonal antibody therapeutics under investigation[36], these results raise the question of whether further suppression early during the course of the disease may be detrimental in the setting of an already suppressed inflammatory response to SARS-CoV-2. That said, it is important to consider that the attenuation of inflammatory pathways we observed may not hold over the course of the disease, especially in severe cases of COVID-19 where the lower and not upper respiratory tract is the primary site of pathology[37–39]. Additional work examining the temporal dynamics of the host response to SARS-CoV-2 in

both the upper and lower airways is needed to address these outstanding questions.

We also leveraged these data to develop an accurate, clinically practical, COVID-19 diagnostic classifier that may help overcome the limitations of direct viral nucleic acid detection. A host transcriptional classifier, utilized alone or in combination with molecular detection of SARS-CoV-2, could reduce both false negative results, for example due to insufficient viral load, or false positive results, for example due to cross-contamination. Nevertheless, the applicability of the classifier developed here could be limited by sample size and incomplete demographic information. Moreover, due to the small proportion of severe COVID-19 cases in our cohort and limited availability of clinical follow-up data, we were unable to examine the important question of whether host transcriptional markers early in the course of disease can provide prognostic value for disease severity and outcomes. Future prospective studies in a larger cohort will be needed to validate our findings, determine the prognostic value of host signatures, and assess the performance of integrated host/ viral diagnostic assays.

## Methods

**Study design, clinical cohort, and ethics statement**. We conducted an observational cohort study of 234 patients with ARIs tested for COVID-19 at the University of California, San Francisco (UCSF) and Zuckerberg San Francisco General Hospital. We evaluated leftover RNA extracted from clinical swab specimens processed at the UCSF Clinical Microbiology Laboratory. The UCSF Institutional Review Board granted a waiver of consent for this study, which was part of a larger ongoing surveillance study of patients with outbreak-associated viral and bacterial infections (UCSF IRB protocol 17-24056).

Inclusion criteria were: (1) status as a patient under investigation for COVID-19, (2) age of 18 years or older, (3) a clinician-ordered test for SARS-CoV-2 was performed between 03/10/2020 and 04/07/2020 using RT-PCR from a nasopharyngeal (NP) swab, obtained with or without an oropharyngeal (OP) swab (Supplementary Table 1), and (4) excess extracted RNA was available for metagenomic sequencing. If more than one sample was collected from a patient ultimately diagnosed with COVID-19, only the first available positive sample was analyzed. Demographic and clinical characteristics were assessed exclusively from each institution's Epic-based electronic health record.

**SARS-CoV-2 detection by clinical PCR**. PCR testing for COVID-19 was carried out in the UCSF Clinical Microbiology Laboratory. Primers targeted either two regions of the SARS-CoV-2 N gene ($n = 153$, 65%), or the E and RNA-dependent RNA polymerase genes ($n = 81$, 35%), plus human RNAse P as a positive control. In all our analyses, we defined patients with COVID-19 as those with a positive SARS-CoV-2 result by PCR.

**Metagenomic sequencing**. To evaluate host gene expression and detect the presence of other viruses, metagenomic next-generation sequencing (mNGS) of RNA was performed on the same specimens subjected to SARS-CoV-2 PCR testing. Following DNase treatment, human cytosolic and mitochondrial ribosomal RNA were depleted using FastSelect (Qiagen). To control for background contamination (see details below), we included negative controls (water and HeLa cell RNA) as well as spike-in RNA standards from the External RNA Controls Consortium (ERCC)[40]. RNA was then fragmented and subjected to a modified metagenomic spiked sequencing primer enrichment (MSSPE) library preparation[41]. In brief, a 1:1 mixture of the NEBNext Ultra II RNA Library Prep (New England Biolabs) random primers and a pool of SARS-CoV-2 primers at 100 µM was used at the first strand synthesis step of the standard RNA-seq library preparation protocol to enrich for reads spanning the length of the SARS-CoV-2 genome. RNA-seq libraries underwent 146 nucleotide paired-end sequencing on an Illumina NovaSeq 6000 instrument. The total number of read-pairs per sample is provided in Supplementary Data 5 (min: 3.1 million, mean: 28.4 million, max: 94 million).

**Quantification of SARS-CoV-2 viral load by mNGS**. All samples were processed through a SARS-CoV-2 reference-based assembly pipeline that involved removing reads likely originating from the human genome or from other viral genomes annotated in RefSeq with Kraken2[42] (v. 2.0.8_beta), and then aligning the remaining reads to the SARS-CoV-2 reference genome MN908947.3 using minimap2[43] (v. 2.17). We calculated SARS-CoV-2 reads-per-million (rpM) using the number of reads that aligned with mapq ≥ 20. For plotting purposes, a value of 0.1 rpM was added to all samples with rpM < 0.1.

**Detection of other pathogenic respiratory viruses by mNGS**. All samples were processed through the IDSeq pipeline[44,45] (v. 4.3), which performs reference-based alignment at both the nucleotide and amino acid level against sequences in the National Center for Biotechnology Information (NCBI) NT and NR databases, respectively, followed by assembly of the reads matching each taxon detected. We further processed the results for viruses with established pathogenicity in the respiratory tract[10]. We evaluated whether one of these viruses was present in a patient sample if it met the following three initial criteria: (i) at least 10 counts mapped to NT sequences, (ii) at least 1 count mapped to NR sequences, (iii) average assembly nucleotide alignment length of at least 70 bp.

Negative control (water and HeLa cell RNA) samples enabled estimation of the number of background reads expected for each virus, which were normalized by input mass as determined by the ratio of sample reads to spike-in ERCC RNA standards[46]. Viruses were then additionally tested for whether the number of sequencing reads aligned to them in the NT database was significantly greater than background. This was done by modeling the number of background reads as a negative binomial distribution, with mean and dispersion fitted on the negative controls. For each batch (sequencing library preparation) and taxon (virus), we estimated the mean parameter of the negative binomial by averaging the read counts across all negative controls after normalizing by ERCCs, slightly regularizing this estimate by including the global average (across all batches) as an additional sample. We estimated a single dispersion parameter across all taxa and batches, using the functions glm.nb() and theta.md() from the R package MASS[47] (v. 7.3-51). We considered a patient to have a pathogenic respiratory virus detected by mNGS if the virus achieved an adjusted $p$-value < 0.05 after Holm–Bonferroni correction for all tests performed in the same sample.

**Host DE analysis**. Following demultiplexing, sequencing reads were pseudo-aligned with kallisto[48] (v. 0.46.1), using the bias correction setting, to an index consisting of all transcripts associated with human protein-coding genes (ENSEMBL v. 99), cytosolic and mitochondrial ribosomal RNA sequences, and the sequences of ERCC RNA standards. Samples retained in the dataset had at least 400,000 estimated counts associated with transcripts of protein-coding genes (min: 400,000, mean: 5.8 million, max: 24.5 million). Gene-level counts were generated from transcript abundance estimates using the R package tximport[49] (v. 1.14) with the lengthScaledTPM method.

Genes were retained for DE analysis if they had at least 10 counts in at least 20% of samples ($n = 15,979$). The analysis was performed with the R package limma[50] (v. 3.42) using quantile normalization, the voom method and the design: ~0 + viral status + gender + age, where viral status was either "SARS-CoV-2", "other virus" or "no virus". The biological gender of all patients was inferred based on chromosome Y gene expression. Age was self-reported, and the age of patients for whom we lacked this information was taken as the mean age of samples with age reported in the respective viral status group. Due to policies prohibiting public disclosure of patient age above 89, we set to 89 the age of two patients in the cohort who were older than 89 so that our analysis can be reproduced with the demographic information we are allowed to make public. DE $p$-values reported in Supplementary Data 2 are based on a moderated $t$-statistic. $p$-values adjusted within each comparison with the method of Benjamini–Hochberg are also reported.

To generate the gene expression heatmap, hierarchical clustering was performed on the union of the top 50 genes (by $p$-value) in each of the pairwise comparisons among the three groups ($n = 121$ genes). Gene counts were subjected to the variance-stabilizing transformation, as implemented in the R package DESeq2[51] (v. 1.26), centered and scaled prior to clustering. The distance measure for rows was based on Pearson correlation and for columns on Euclidean distance. Ward's criterion (ward.D2) was the agglomeration method for both rows and columns.

The independent NP swab mNGS dataset was generated as part of a study that did not address the comparison between COVID-19 and other viral ARIs[16,27], and was re-analyzed here for this purpose. The dataset was filtered to retain samples with at least 5 million counts and genes with at least 32 counts in at least 10% of samples. A DE analysis was then performed using limma, with quantile normalization and the voom method, between patients who tested negative for SARS-CoV-2 by RT-PCR ($n = 308$), patients who tested positive for SARS-CoV-2 ($n = 166$), and patients who tested negative for SARS-CoV-2 but had other respiratory viruses detected ($n = 79$). No covariates were included in the analysis.

**Gene set enrichment analysis**. GSEA was performed on REACTOME[52] pathways with a minimum size of 10 genes and a maximum size of 1500 genes using the fgseaMultilevel function in the R package fgsea[53] (v. 1.13.5), which calculates $p$-values based on an adaptive, multilevel splitting Monte Carlo scheme. Genes included in each pairwise comparison were those with a Benjamini–Hochberg adjusted $p$-value < 0.1 in the respective DE analysis, pre-ranked by fold-change. The gene sets shown in Fig. 1b were manually selected to reduce redundancy and highlight diverse biological functions from among those with a Benjamini–Hochberg adjusted $p$-value < 0.05 in at least one of the comparisons (i) SARS-CoV-2 vs. no virus, and (ii) other virus vs. no virus. The gene sets shown in Supplementary Fig. 2b were similarly selected from among those with an adjusted $p$-value < 0.05 in the direct comparison of SARS-CoV-2 vs. other virus. Full GSEA results are provided as Supplementary Data 3.

**Regression of gene counts against viral load**. We performed robust regression of the limma-generated quantile normalized gene counts ($\log_2$ scale) against $\log_{10}(\text{rpM})$ of SARS-CoV-2 for all genes with a Benjamini–Hochberg adjusted $p$-value $< 0.001$ in either the DE analysis of SARS-CoV-2 vs. no virus, or SARS-CoV-2 vs. other virus ($n = 3,636$ genes). The samples included in the regression analysis were those in the SARS-CoV-2 patient group with $\log_{10}(\text{rpM}) \geq 0$ ($n = 82$ samples).

The analysis was performed using the R package robustbase[54] (v. 0.93.6), which implements MM-type estimators for linear regression[55,56], using the KS2014 setting and the model: quantile normalized counts ($\log_2$ scale) ~ gender + age + $\log_{10}(\text{rpM})$. Model predictions based on the marginal effects of $\log_{10}(\text{rpM})$ were generated using the R package ggeffects (v. 0.14.3) and used for display in the individual gene plots. Error bands represent normal distribution 95% confidence intervals around each prediction. Reported $p$-values for significance of the difference of the regression coefficient from 0 are based on a $t$-statistic and Benjamini–Hochberg adjusted. Reported $R^2$ values represent the adjusted robust coefficient of determination[57].

**In silico analysis of cell-type proportions**. Cell-type proportions were estimated from bulk host transcriptome data using the CIBERSORT X algorithm[58]. We used the human lung cell atlas dataset[59] to derive the single-cell signatures. The cell types estimated with this reference cover all expected cell types in the airway. The estimated proportions were compared between the three patient groups using a Mann–Whitney–Wilcoxon test (two-sided) with Bonferroni correction.

**Classifier construction**. We built sparse classifiers for COVID-19 status using a combined lasso and random forest approach. For feature selection, we used the logistic lasso (as implemented in the R package glmnet[60], v. 4.0-2), and then trained random forests on the selected features (using the R package random-Forest[61], v. 4.6-14). We used fivefold cross-validation to evaluate model error. For each train-test split, we used a nested cross-validation within the training set to select the lasso tuning parameter, using glmnet's default "1se" rule (selecting the highest lambda whose error was within one standard error of the optimum). For the 10-gene and 3-gene models, we selected the smallest lambda with 10 (respectively, 3) nonzero coefficients, and evaluated the cross-validation errors at that value of lambda. For the random forest, we used 100,000 trees, and left all tuning parameters at their defaults. For the initial input features (before selection), we used gene counts with a variance-stabilizing transform derived from the training set only, using the R package DESeq2. Classifier performance was benchmarked against the SARS-CoV-2 PCR results.

**Reporting summary**. Further information on research design is available in the Nature Research Reporting Summary linked to this article.

## Data availability

Human gene counts and metadata for the samples generated in this study can be obtained at: https://github.com/czbiohub/covid19-transcriptomics-pathogenesis-diagnostics-results. Gene counts have also been deposited under NCBI GEO accession GSE156063. IDSeq metagenomic analysis reports are available at https://idseq.net/ under project name "covid19_transcriptomics_pathogenesis_diagnostics". Raw mNGS FASTQ files, subtracted of human-mapping reads for privacy reasons, are available under NCBI BioProject accession PRJNA633853. The independent NP swab mNGS dataset we re-analyzed can be obtained according to the data availability statement in the original publication[27]. The published human lung single-cell datasets[59] used for cell-type proportions analysis can be obtained through Synapse under accessions syn21560510 and syn21560511.

## Code availability

Code for parsing the IDSeq reports and for the differential expression analyses, cell-type proportions analysis and gene expression classifiers is available at: https://github.com/czbiohub/covid19-transcriptomics-pathogenesis-diagnostics-results.

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

## Acknowledgements

We thank Maíra Phelps for clinical research coordination efforts. We also thank Steve Miller, the UCSF Clinical Microbiology Laboratory and the CLIAHUB team at the Chan Zuckerberg Biohub for assistance with sample acquisition. This study was supported by the Chan Zuckerberg Biohub, the Chan Zuckerberg Initiative, the National Heart, Lung, and Blood Institute (1K23HL138461-01A1), and philanthropic contributions from Mark and Carrie Casey, Julia and Kevin Hartz, Carl Kawaja and Wendy Holcombe, Eric Keisman and Linda Nevin, Martin and Leesa Romo, Diana Wagner, Jerry Yang and Akiko Yamazaki, and Three Sisters Foundation.

## Author contributions

C.L., A.K., E.M., and J.K. conceived and designed the study. A.K., N.N., G.C., A.M.D., G.R.K., P.H.S., M.T., and S.A.M. oversaw or performed sample processing, library preparation, and sequencing. C.L., S.J.S., K.N.K., P.A.P., K.R., L.M.L., and N.S. performed metadata collation or clinical chart review. E.M., J.K., A.O.P., C.L., K.R., and S.L.H. performed data analysis. S.A.C. contributed to data interpretation. E.M., J.K., and A.O.P. generated data visualizations. C.S.C., J.L.D., and J.M.B. provided guidance, advice, and comments on the study design and manuscript. C.L., E.M., J.K., and S.A.C. wrote the manuscript with input from all authors.

## Competing interests

The authors declare no competing interests.
