## [Peer Review File · Nature Communications]

Reviewers' Comments:

Reviewer #1:

Remarks to the Author:

The authors performed metagenomic sequencing to study the viral and host transcriptional profiles of three cohorts: COVID-19 patient, other viral or non-viral acute respiratory illnesses (ARIs). This paper draws conclusions mainly from the following perspectives: 1. Characteristics of the host immune response unique to COVID-19 patients; 2. The feasibility of using host gene expression for COVID-19 diagnostic.

Overall, this paper presents a solid study. Here are some specific comments:

- A large number of individuals were included in whom no virus was detected, despite the presence of an acute respiratory illness (ARI). What do the authors believe is the etiology of the ARI in these patients? Did the authors consider the possibility that a proportion of these presumably "no virus" cases tested false-negative for CoV2? Was imaging performed in these patients for further investigation? If CoV2 testing in some of these "no virus" cases was false-negative, the subsequent development of diagnostic classifiers would be of limited help.
- A weakness of this study is that transcriptional profiles were not tested for possible prognostic significance. Was there any correlation between transcriptional signatures and disease progression/disease outcome parameters? Using transcriptional data for risk stratification of CoV2 patients for development of severe/critical illness would (in my opinion) be more important than trying to use them as diagnostic tools.
- Downregulation of innate immune recognition and immune signaling pathways in CoV2 patients (Figure 1B) is potentially the most important finding of this study. Mechanistic studies would be very helpful to strengthen this study: How is CoV2 RNA being sensed and recognized by innate immune receptors? How is the IFN response being induced? How effective is the IFN response in controlling viral replication?
- While of interest, the inferred recruitment of immune cells to NP mucosa through biocomputational analysis is a bit inconclusive – what message do the authors wish to communicate with these findings?
- For Figure 1D, Figure S2A and Figure S2C, there are a few data points from COVID-19 patients with a $\log_{10}(\text{rpM}) < 0$. Are they included in the regression analysis of gene counts against viral abundance? If not, they may impact the results.

Reviewer #2:

Remarks to the Author:

This comparison of transcriptomes in COVID-19 and other acute respiratory illnesses is an important contribution to the research on the new pandemic disease. The study and the analysis are also very well performed and presented. The reviewer acknowledges that covariate-adjustments and nested-cross validation was used.

Here are some comments the authors could consider to improve their manuscript:

- 1) Besides accuracy, sensitivity and specificity, positive and negative predictive values from the cross validation should be reported.
- 2) For metagenomics analysis, the authors map only against the SARS-CoV-2 genome. Just as an idea: would it be possible, here, to map against all other available viral genomes in order to obtain a full view on the virome in the three patient groups?
- 3) In the Supplementary Methods, the first subchapter is named "...ethics statement", but I don't see this statement on informed study participation there, or about a vote on the study from an ethics committee.
- 4) The authors model age and gender as covariates in the differential expression analysis, although

there are no large differences between the groups (Table S1). Is this maybe an overfitting? Other patient characteristics show more differences between the three groups.

5) Gene-set-enrichment analysis according to Subramanian et al. does not need a threshold for the gene list. However, the authors describe threshold in the method description.

Reviewer #3:

Remarks to the Author:

In their manuscript, Mick et al describe gene expression changes in SARS-CoV-2 infected patients. The authors collected swab samples from SARS-CoV-2 infected patients and compared them to swab samples from patients infected with other viruses or patients with no acute viral infections but acute respiratory illness. Their analysis shows SARS-CoV-2 specific gene host immune response signatures and specific pattern of SARS-CoV-2 for cellular markers. From these data, they generated classifiers to distinguish SARS-CoV-2 infections from other viral or non-viral respiratory illnesses. They claimed that this might be useful to increase sensitivity for positively testing patients for a SARS-CoV-2 infection. The authors analyzed a total of 94 SARS patients, 41 viral infections, and 103 non-viral respiratory illnesses. These are very good group sizes for such an analysis. The approaches and analysis methods that were used in the manuscript are all appropriate and technically well performed. A few minor questions remain, see below. The scripts and processed data have been deposited in a public repository. The identification of classifiers to distinguish SARS infections from other viral infections is very promising but needs to be validated in clinical settings. The limitations of the study are briefly mentioned.

The manuscript is acceptable for publication after major revisions.

Major points

Page 12, line 210: they describe that they used swab specimens. Excluded where swabs from oropharyngeal regions. It is not clear from that description what kind of swabs were actually used for their analysis. Please specify.

Page 17, line 340: I guess positivity of the PCR was used for classification. Mentioning the gold standard is a bit confusing.

Page 12, line 224: please explain abbreviations of NP and OP.

Page 13, line 242: what was the total number of reads per sample?

I did not find a reference for the deposition of the raw sequence reads. These should also be available to the scientific community.

Minor points

I would appreciate if in the abstract not only the total number of samples but also the specific numbers for the different groups was reported.

A limitation that should also be mentioned is that the analysis only studied the upper respiratory tract but not the lungs which are the most critical infected tissues for severe disease.

In a supplement, I would appreciate illustrations for individual genes in a boxplot with individual value points for IFI27, IFI6, IL1R2, ACE2, IFI44L, OAS, IL1B, TRO, EIF4A2 genes that also show the expression levels separately for the different viruses in the non-SARS group. Are they all different or do some these viruses show similar patterns like SARS?

What does an increase in the cellular proportion of goblet cells mean? Sloughing off of cells? Please comment.

Reviewer #1 (Remarks to the Author):

The authors performed metagenomic sequencing to study the viral and host transcriptional profiles of three cohorts: COVID-19 patient, other viral or non-viral acute respiratory illnesses (ARIs). This paper draws conclusions mainly from the following perspectives: 1. Characteristics of the host immune response unique to COVID-19 patients; 2. The feasibility of using host gene expression for COVID-19 diagnostic.

Overall, this paper presents a solid study. Here are some specific comments:

- A large number of individuals were included in whom no virus was detected, despite the presence of an acute respiratory illness (ARI). What do the authors believe is the etiology of the ARI in these patients? Did the authors consider the possibility that a proportion of these presumably "no virus" cases tested false-negative for CoV2? Was imaging performed in these patients for further investigation? If CoV2 testing in some of these "no virus" cases was false-negative, the subsequent development of diagnostic classifiers would be of limited help.

We thank the reviewer for raising this important point. We considered the possibility that patients in the "no virus" group might have had a false negative test for SARS-CoV-2. However, given that each sample effectively underwent 2 tests (initially by PCR and then again by mNGS), 2 false negatives would have had to happen, which is less likely.

To address the etiology of ARI in "no virus" patients, three physicians on our team performed a chart review of the patients in this group and we have compiled the clinical diagnoses in new **Supplementary Data 1**. Many patients received plausible alternative diagnoses, including bacterial pneumonia and non-infectious lung and airway conditions (e.g., allergy, asthma, COPD exacerbations) although a definitive etiology was not determined in every case, as is usual with ARIs. Chest x-rays were available to clinicians making these diagnoses in most cases, and we also included the imaging results in the new table. We have revised the manuscript as follows:

Lines 25-27: "Diagnoses in the latter group included bacterial pneumonia and non-infectious lung and airway conditions, though a definitive etiology could not be determined in every case (Supplementary Data 1)."

As part of the chart review, we noticed that 3 "no virus" patients had a positive viral test of some kind within 7 days of collection of the swab included in the study. To ensure patient groups are as clearly delineated as possible, we have now entirely removed these 3 samples from the study and additionally removed the 1 sample from the "SARS-CoV-2" group we previously indicated had a co-infection with another virus. Thus, the revision now reflects the absence of 4 samples and the cohort composition is: n=93 "SARS-CoV-2", n=41 "other virus", n=100 "no virus". While quantitative results naturally shifted somewhat, the principal conclusions were unaffected.

A final source of reassurance in this regard is the host classifier itself. Under the reasonable assumption that, at a minimum, the clear majority of “no virus” samples were not falsely negative for SARS-CoV-2 – then the per-sample predictions produced by the classifier can provide a sense of whether a few “no virus” samples may have harbored occult infection. In fact, that is precisely one of the applications we envision for a host classifier. In practice, only 2 “no virus” samples received a 50% chance or higher of having been falsely negative.

- A weakness of this study is that transcriptional profiles were not tested for possible prognostic significance. Was there any correlation between transcriptional signatures and disease progression/disease outcome parameters? Using transcriptional data for risk stratification of CoV2 patients for development of severe/critical illness would (in my opinion) be more important than trying to use them as diagnostic tools.

We absolutely agree that identifying transcriptional profiles early in the course of disease correlating with prognosis is valuable and needed. Unfortunately, the proportion of COVID-19 patients hospitalized or in the ICU at the time of testing in our cohort was small, and due to our study protocol which leveraged leftover swab specimens from our hospital’s clinical microbiology laboratory, we did not have access to complete outcomes data, in particular follow up assessment of disease severity or any subsequent admission to other hospitals. As such, we were insufficiently powered to assess prognostic significance of transcriptional profiles on key outcomes such as hospitalization, ICU admission and mortality. We have now indicated in the Discussion section that this represents a limitation of our study and that future studies designed to assess prognostic significance are needed:

Lines 158-163: “Moreover, due to the small proportion of severe COVID-19 cases in our cohort and limited availability of clinical follow-up data, we were unable to examine the important question of whether host transcriptional markers early in the course of disease can provide prognostic value for disease severity and outcomes. Future prospective studies in a larger cohort will be needed to validate our findings, determine the prognostic value of host signatures, and assess the performance of integrated host/viral diagnostic assays.”

We would also like to note that we have designed and begun enrollment in a new prospective study that includes longitudinal collection of blood, upper and lower respiratory tract specimens during hospitalization and following discharge.

- Downregulation of innate immune recognition and immune signaling pathways in CoV2 patients (Figure 1B) is potentially the most important finding of this study. Mechanistic studies would be very helpful to strengthen this study: How is CoV2 RNA being sensed and recognized by innate immune receptors? How is the IFN response being induced? How effective is the IFN response in controlling viral replication?

We agree that attenuation of innate immune signaling pathways in the upper airway during SARS-CoV-2 infection, possibly mediated by the virus, is the most significant contribution of this study, as it may provide an explanation for how the virus achieves high titers (and therefore high transmissibility) prior to the onset of symptoms. In support of this connection, we now cite a human challenge study with influenza virus that specifically correlated lack of activation of the

IL-1 and NLRP3 inflammasome pathways with an asymptomatic course of infection (PMID 21901105).

We absolutely agree that mechanistic studies will be a critical next step. However, we believe such experiments are beyond the scope of the current brief report, which aims to rapidly communicate these important findings to the scientific community. Moreover, the significant differences between the signals we and others observed in patients and recently published results from cell culture or even animal models of SARS-CoV-2 infection (for example, with respect to the interferon response, PMID 32416070) demonstrate that achieving physiologically relevant mechanistic insight will pose a complex, long-term challenge given the likely involvement of species-specific factors and cell non-autonomous effects.

We now mention some of these ideas in the Discussion section:

Lines 133-142: "The blunted activation of these pro-inflammatory pathways in the upper airway carries two important implications. First, it suggests a potential mechanism underlying the observation of high viral titers in the upper airway prior to symptom onset in infected individuals, which poses a major challenge to preventing viral transmission in the COVID-19 pandemic and distinguishes it from the SARS outbreak of 2002-2003. Intriguingly, a human challenge study with influenza virus reported that blunted activation of the IL-1 and NLRP3 inflammasome pathways was specifically associated with an asymptomatic course of infection, supporting this connection. Influenza was indeed the only other virus that showed a degree of similarity to SARS-CoV-2 in this regard in our data. Future work is needed to identify potential SARS-CoV-2 factors responsible for this immune attenuation and to clarify the underlying biological mechanisms."

- While of interest, the inferred recruitment of immune cells to NP mucosa through biocomputational analysis is a bit inconclusive - what message do the authors wish to communicate with these findings?

Variations in cell type composition contribute substantially to variations in bulk tissue gene expression. Cell type deconvolution analysis can thus provide a layer of context to better understand the observed differential expression at the gene and pathway level. For example, several of the genes most downregulated in COVID-19 samples compared to other viral ARIs are those involved in inflammasome activation. While inflammasome-associated mediators are highly expressed in monocytes and tissue macrophages, the inflammasome machinery can also be activated in other airway cells (e.g., epithelial, dendritic and B cells). However, the deconvolution analysis specifically revealed decreased proportions of infiltrating macrophages, as well as the neutrophils that activate them, in COVID-19 samples compared to other viral ARIs, which was not the case for epithelial, dendritic or B cells. Thus, these findings support the idea that blunted recruitment and/or activation of macrophages may at least partly explain the differences in inflammasome gene expression. To emphasize this connection, we now state:

Abstract: "...the IL-1 and NLRP3 inflammasome pathways were markedly less responsive to SARS-CoV-2, commensurate with a signature of diminished neutrophil and macrophage recruitment..."

Lines 72-82: "Strikingly, while patients with other viral ARIs exhibited significantly increased fractions of monocytes/macrophages and neutrophils in the upper airway compared to non-viral ARIs, this was not the case for those infected with SARS-CoV-2 (Fig. 1c, Supplementary Fig. 4). SARS-CoV-2 infection increased proportions of dendritic

and B-cells more than other viruses, while other viruses decreased proportions of ciliated cells and goblet cells. These results closely aligned with the GSEA findings and suggested that the diminished innate immune responses observed in COVID-19 patients were coupled to blunted recruitment and activation of pro-inflammatory macrophages in the airway microenvironment. Further supporting these findings, we found that the gene most depressed in expression in COVID-19 patients compared to those with other viral ARIs was IL1B, which encodes a pro-inflammatory cytokine produced by the inflammasome complex, particularly in macrophages (Fig. 1d, Supplementary Data 2).”

- For Figure 1D, Figure S2A and Figure S2C, there are a few data points from COVID-19 patients with a $\log_{10}(\text{rpM}) < 0$. Are they included in the regression analysis of gene counts against viral abundance? If not, they may impact the results.

We appreciate the opportunity to clarify. The regression analysis was indeed applied to samples from the “SARS-CoV-2” group with $\log_{10}(\text{rpM}) \geq 0$, as indicated in the figure legend, by the dotted line on the x-axis and in the Methods section.

The rationale for this is that the regression seeks to quantify the linear relationship between gene expression and viral load, but the quantitative measurement of viral load by mNGS for samples with $\log_{10}(\text{rpM}) < 0$ is no longer accurate and would thus simply inject noise into the analysis. Given that the vast majority of “SARS-CoV-2” samples met the threshold (82/93 samples in the revision), and that these samples spanned several orders of magnitude of viral load, we believe the exclusion of the low abundance samples does not affect our ability to distinguish genes with strong linear relationships (e.g., OASL in Supplementary Fig. 2) from those with weak ones (e.g., IFI27 in Supplementary Fig. 2).

We note that while it would not be informative to directly include the low abundance samples in the regression analysis, they still provide useful insight that readers can glean from the plots and data tables. For example, for some of the interferon-inducible genes shown in Supplementary Fig. 2, there is a threshold of viral abundance at which they become strongly induced in an almost stepwise fashion (IFI6, IFI44L, IIF27) whereas for other genes (IRF7, OASL) such stepwise behavior is absent.

Reviewer #2 (Remarks to the Author):

This comparison of transcriptomes in COVID-19 and other acute respiratory illnesses is an important contribution to the research on the new pandemic disease. The study and the analysis are also very well performed and presented. The reviewer acknowledges that covariate-adjustments and nested-cross validation was used.

Here are some comments the authors could consider to improve their manuscript:

1) Besides accuracy, sensitivity and specificity, positive and negative predictive values from the cross validation should be reported.

We appreciate the reviewer’s suggestion and have now updated **Supplementary Table 3** with the sensitivity, specificity, positive predictive value (PPV) and negative predictive value (NPV) of

the classifier i) with respect to the overall cohort, ii) with respect to the “no virus” group, and iii) with respect to the “other virus” group.

2) For metagenomics analysis, the authors map only against the SARS-CoV-2 genome. Just as an idea: would it be possible, here, to map against all other available viral genomes in order to obtain a full view on the virome in the three patient groups?

We appreciate the opportunity to clarify – that is in fact what we have done. As indicated in the Methods section, we mapped all samples against the NCBI NT and NR databases using the IDSeq metagenomic analysis pipeline, and then applied statistical tests to determine whether any other pathogenic respiratory virus (beyond SARS-CoV-2) was present in the samples. This was the basis for partitioning samples that tested negative for SARS-CoV-2 by PCR into either the “no virus” group or the “other virus” group.

The only purpose for which we mapped exclusively against the SARS-CoV-2 genome was to determine SARS-CoV-2 viral load, measured in reads-per-million (rpM). The reason for doing so separately from the IDSeq pipeline was that the latter applies certain heuristics for computational efficiency when mapping against all of the NT/NR database (for example, subsampling of reads) that we wished to avoid when quantifying the abundance of one specific virus of interest. We note that a k-mer based filtering step is still applied to exclude reads likely originating from the human genome or from other viruses prior to mapping to the SARS-CoV-2 genome for rpM purposes.

3) In the Supplementary Methods, the first subchapter is named “...ethics statement”, but I don't see this statement on informed study participation there, or about a vote on the study from an ethics committee.

We appreciate the reviewer highlighting the need to clarify. The UCSF Institutional Review Board granted a waiver of consent for this study, which was part of a larger ongoing surveillance study of patients with outbreak-associated viral and bacterial infections. We now state:

Lines 169-172: “The UCSF Institutional Review Board granted a waiver of consent for this study, which was part of a larger ongoing surveillance study of patients with outbreak-associated viral and bacterial infections (UCSF IRB protocol 17-24056).”

4) The authors model age and gender as covariates in the differential expression analysis, although there are no large differences between the groups (Table S1). Is this maybe an overfitting? Other patient characteristics show more differences between the three groups.

We made the decision to include age and gender covariates in the differential expression analysis *a-priori* in our experimental design both because it is standard practice in analysis of (sufficiently large) human patient cohorts and because of reports of possible differences related to these variables in COVID-19. It is true that the covariates are reasonably well-balanced in the eventual cohort, and furthermore we did not observe strong differential effects of SARS-CoV-2 infection based on them, and so their impact on the results is small. The inclusion of these covariates does not, however, act to inflate the significance of association of any gene with the main variable of interest, i.e. viral status, and is in fact conservative in that regard. We do not

believe we were sufficiently powered, or that there was strong biological motivation, to include other covariates.

In the process of carefully considering this comment, we made a few small adjustments in the DE analysis. First, due to the prohibition on publicly disclosing patient age >89, we have now censored the age of patients in our analysis at 89 so we can report results that would be fully reproducible with the metadata we are allowed to make public. In practice, this only affects 2 patients in the cohort and the results are virtually identical. Second, we previously inferred the biological gender of patients using chromosome Y gene expression only if the self-reported gender was missing or ambiguous. In the interest of consistency, we have now changed this to infer the gender from the data for all patients, which uncovered a small number of patients with discordant gender compared to the annotation in the medical record. Finally, we have removed the sequencing batch variable since we did not detect a meaningful batch effect and its inclusion unnecessarily limited the set of samples that were directly compared to one another. None of these changes altered the principal conclusions of the study.

5) Gene-set-enrichment analysis according to Subramanian et al. does not need a threshold for the gene list. However, the authors describe threshold in the method description.

We appreciate the reviewer's point. Gene set enrichment analysis is often performed using different approaches for ranking and filtering the genes, prior to calculating the enrichment statistics as developed in Subramanian et al. We chose to rank based on fold-change rather than based on a measure of statistical significance, as it has the most direct bearing on biological function. When ranking based on effect size, it becomes necessary to apply at least a light filter for statistical significance to avoid spurious results. However, we also applied a threshold on the fold-change, which we have now removed following the reviewer's comment. We agree setting a fold-change threshold may not be advisable since it can obscure small effects across a large number of genes in a pathway, which is what GSEA was specifically designed to uncover. Indeed, when repeating the analysis without the fold-change threshold, we observed downregulation of the olfactory receptor pathway in SARS-CoV-2, which could be related to the loss of sense of smell reported in COVID-19. In all other respects, our previous conclusions were unaffected.

Reviewer #3 (Remarks to the Author):

In their manuscript, Mick et al describe gene expression changes in SARS-CoV-2 infected patients. The authors collected swab samples from SARS-CoV-2 infected patients and compared them to swab samples from patients infected with other viruses or patients with no acute viral infections but acute respiratory illness. Their analysis shows SARS-CoV-2 specific gene host immune response signatures and specific pattern of SARS-CoV-2 for cellular markers. From these data, they generated classifiers to distinguish SARS-CoV-2 infections from other viral or non-viral respiratory illnesses. They claimed that this might be useful to increase sensitivity for positively testing patients for a SARS-CoV-2 infection.

The authors analyzed a total of 94 SARS patients, 41 viral infections, and 103 non-viral respiratory illnesses. These are very good group sizes for such an analysis. The approaches and analysis methods that were used in the manuscript are all appropriate and technically well

performed. A few minor questions remain, see below. The scripts and processed data have been deposited in a public repository. The identification of classifiers to distinguish SARS infections from other viral infections is very promising but needs to be validated in clinical settings. The limitations of the study are briefly mentioned. The manuscript is acceptable for publication after major revisions.

Major points

Page 12, line 210: they describe that they used swab specimens. Excluded where swabs from oropharyngeal regions. It is not clear from that description what kind of swabs were actually used for their analysis. Please specify.

We appreciate the reviewer's input. The patients included in the study were tested for SARS-CoV-2 using a nasopharyngeal (NP) swab, obtained with or without an additional oropharyngeal (OP) swab. We did not include patients whose only available swab was an OP swab. We have listed the distribution of swab types in Supplementary Table 1. We have also clarified in the text as follows:

Lines 173-177: "Inclusion criteria were... 3) a clinician-ordered test for SARS-CoV-2 was performed between 03/10/2020 and 04/07/2020 using reverse transcription polymerase chain reaction (RT-PCR) from a nasopharyngeal (NP) swab, obtained with or without an oropharyngeal (OP) swab (Supplementary Table 1) ..."

Page 17, line 340: I guess positivity of the PCR was used for classification. Mentioning the gold standard is a bit confusing.

We have incorporated the reviewer's feedback and revised the text as follows:

Line 296: "Classifier performance was benchmarked against the SARS-CoV-2 PCR results."

Page 12, line 224: please explain abbreviations of NP and OP.

We have incorporated this feedback and have now spelled out nasopharyngeal (NP) and oropharyngeal (OP) in lines 15, 175-176 and in the legend of Supplementary Table 1.

Page 13, line 242: what was the total number of reads per sample? I did not find a reference for the deposition of the raw sequence reads. These should also be available to the scientific community.

We thank the reviewer for this input. We have now added a new **Supplementary Data 5** that includes the total number of reads obtained for each sample and the number of reads mapping to transcripts of protein-coding genes. We have deposited the raw sequencing files, following subtraction of human mapping reads, in the publicly accessible NCBI SRA database under Bioproject PRJNA633853. Host gene counts are now deposited in the NCBI GEO database under accession GSE156063. This is in accordance with our IRB protocol allowing waiver of consent and release of only non-identifiable human transcriptomic data. We have clarified this in the text:

Line 201: "Total reads per sample are provided in Supplementary Data 5."

Lines 300-303: "Original mNGS FASTQ files, subtracted of human-mapping reads for privacy reasons, are available under NCBI BioProject accession PRJNA633853. Gene counts are also available under NCBI GEO accession GSE156063."

Minor points

I would appreciate if in the abstract not only the total number of samples but also the specific numbers for the different groups was reported.

We appreciate this suggestion and have now added the number of samples from each group to the abstract.

A limitation that should also be mentioned is that the analysis only studied the upper respiratory tract but not the lungs which are the most critical infected tissues for severe disease.

We agree and have now emphasized this in the Discussion section, as suggested:

Lines 146-151: "That said, it is important to consider that the attenuation of inflammatory pathways we observed may not hold over the course of disease, especially in severe cases of COVID-19 where the lower and not upper respiratory tract is the primary site of pathology. Additional work examining the temporal dynamics of host response to SARS-CoV-2 in both the upper and lower airways is needed to address these outstanding questions."

In a supplement, I would appreciate illustrations for individual genes in a boxplot with individual value points for IFI27, IFI6, IL1R2, ACE2, IFI44L, OAS, IL1B, TRO, EIF4A2 genes that also show the expression levels separately for the different viruses in the non-SARS group. Are they all different or do some these viruses show similar patterns like SARS?

We appreciate this suggestion and have now added **Supplementary Fig. 3** with boxplots comparing the expression of these and a few more genes across the different types of other viruses identified in our cohort. Overall, the differences between SARS-CoV-2 and other viruses were quite consistent across virus types. However, we did observe that influenza was perhaps the only other virus that did not consistently activate the IL-1 and NLRP3 inflammasome pathways, similarly to SARS-CoV-2, and we now comment on this in the text:

Lines 86-89: "We note that the muted response of the IL-1 and NLRP3 inflammasome pathways to SARS-CoV-2 infection appeared to distinguish it from most other pathogenic respiratory viruses in our cohort, including common cold coronaviruses, with the possible exception of influenza (Supplementary Fig. 3b)."

What does an increase in the cellular proportion of goblet cells mean? Sloughing off of cells? Please comment.

We appreciate this comment and upon further consideration of the data, we believe the effect size of goblet cell hyperplasia in COVID-19 patients is in fact quite small. As such, we have now de-emphasized this observation in the text.

Reviewers' Comments:

Reviewer #1:

Remarks to the Author:

The authors have improved the manuscript and adequately addressed/discussed my prior concerns. I have no further questions.

Reviewer #2:

Remarks to the Author:

The authors have revised and improved their manuscript according to all points risen by myself and by two other reviewers.

Reviewer #3:

Remarks to the Author:

The authors satisfactorily responded to all comments.
The data have been deposited at GEO.

One minor point:

In the results or M&M, please report the mean, min and max number of reads per sample.

The article is now acceptable for publication.

Reviewer #1 (Remarks to the Author):

The authors have improved the manuscript and adequately addressed/discussed my prior concerns. I have no further questions.

Thank you.

Reviewer #2 (Remarks to the Author):

The authors have revised and improved their manuscript according to all points risen by myself and by two other reviewers.

Thank you.

Reviewer #3 (Remarks to the Author):

The authors satisfactorily responded to all comments.
The data have been deposited at GEO.

One minor point:

In the results or M&M, please report the mean, min and max number of reads per sample.

The article is now acceptable for publication.

Thank you. We have added the min, mean and max of the total sequencing reads per sample as well as the counts associated with protein coding genes in the Methods section.